# Boron Neutron Capture Therapy Followed by Image-Guided Intensity-Modulated Radiotherapy for Locally Recurrent Head and Neck Cancer: A Prospective Phase I/II Trial

**DOI:** 10.3390/cancers15102762

**Published:** 2023-05-15

**Authors:** Ling-Wei Wang, Yen-Wan Hsueh Liu, Pen-Yuan Chu, Hong-Ming Liu, Jinn-Jer Peir, Ko-Han Lin, Wen-Sheng Huang, Wen-Liang Lo, Jia-Cheng Lee, Tzung-Yi Lin, Yu-Ming Liu, Sang-Hue Yen

**Affiliations:** 1Department of Heavy Ion and Radiation Oncology, Taipei Veterans General Hospital, No. 201, Section 2, Shih-Pai Road, Taipei 11217, Taiwan; 2School of Medicine, National Yang-Ming Chiao Tung University, No. 155, Section 2, Li-Nong Street, Taipei 112304, Taiwan; 3Heron Neutron Medical Corporation, No. 66-2, Shengyi 5th Road, Zhubei City 30261, Taiwan; yvonne.hsueh@heron-neutron.com (Y.-W.H.L.);; 4Department of Otolaryngology, Taipei Veterans General Hospital, No. 201, Section 2, Shih-Pai Road, Taipei 11217, Taiwan; 5Nuclear Science and Technology Development Center, National Tsing Hua University, No. 101, Sect 2, Kuang Fu Road, Hsinchu 30013, Taiwan; 6Department of Nuclear Medicine, Taipei Veterans General Hospital, No. 201, Section 2, Shih-Pai Road, Taipei 11217, Taiwan; 7Department of Nuclear Medicine, Cheng Hsin General Hospital, No. 45, Cheng Hsin Street, Taipei 11220, Taiwan; 8Department of Stomatology, Taipei Veterans General Hospital, No. 201, Section 2, Shih-Pai Road, Taipei 11217, Taiwan; 9School of Dentistry, National Yang-Ming Chiao Tung University, No. 155, Section 2, Li-Nong Street, Taipei 112304, Taiwan; 10Division of Radiation Oncology, Wan Fang Hospital, Taipei Medical University, No. 111, Section 3, Shing-Long Road, Taipei 116, Taiwan

**Keywords:** head and neck cancer, boron neutron capture therapy, boronophenylalanine, image-guided intensity-modulated radiotherapy

## Abstract

**Simple Summary:**

Boron neutron capture therapy (BNCT) is a target radiotherapy and image-guided intensity-modulated radiotherapy (IG-IMRT) that has been used for recurrent head and neck cancer. To procure better results, the current study designed to combine both as salvage treatment for this challenging problem. For the 14 patients enrolled, a high response rate and low incidence of grade 4 toxicity were seen at follow-up. Further local recurrence was the main cause of failure in spite of the larger radiation volumes and extra dose given with IG-IMRT after BNCT than with BNCT alone. Further revision of this protocol is suggested to improve our results.

**Abstract:**

Background: This trial investigated the efficacy and safety of salvage boron neutron capture therapy (BNCT) combined with image-guided intensity-modulated radiotherapy (IG-IMRT) for recurrent head and neck cancer after prior radiotherapy (RT). Methods: BNCT was administered using an intravenous boronophenylalanine–fructose complex (500 mg/kg) in a single fraction; multifractionated IG-IMRT was administered 28 days after BNCT. For BNCT, the mucosa served as the dose-limiting organ. For IG-IMRT, the clinical target volume (CTV) and the planning target volume (PTV) were generated according to the post-BNCT gross tumor volume (GTV) with chosen margins. Results: This trial enrolled 14 patients, and 12 patients received combined treatment. The median BNCT average dose for the GTV was 21.6 Gy-Eq, and the median IG-IMRT dose for the PTV was 46.8 Gy/26 fractions. After a median (range) follow-up period of 11.8 (3.6 to 53.2) months, five patients had a complete response and four had a partial response. One patient had grade 4 laryngeal edema; another patient had a grade 4 hemorrhage. Most tumor progression occurred within or adjacent to the CTV. The 1-year overall survival and local progression-free survival rates were 56% and 21%, respectively. Conclusion: Despite the high response rate (64%) of this trial, there was a high incidence of in-field and marginal failure with this approach. Future studies combining BNCT with modalities other than radiation may be tried.

## 1. Background

Head and neck cancer (HNC), particularly oral cancer, constitutes an endemic disease in Taiwan. Despite advances in surgery, radiotherapy (RT), and chemotherapy techniques, local recurrence of HNC after multidisciplinary treatment is not uncommon. Several studies have explored reirradiation for recurrent HNC, but the corresponding complication rates remained high [1,2] and success rates were low [3,4]. Consequently, identifying additional treatment options for patients with recurrent HNC is imperative. The rationale underlying boron neutron capture therapy (BNCT) is a nuclear capture reaction that occurs when low-energy thermal neutrons collide with nonradioactive ^10^B. High-energy alpha particles (^4^He^2+^) and ^7^Li nuclei were produced afterwards [5]. Because the path lengths of such particles and nuclei are approximately equal to one cell diameter (<10 μm), the therapeutic effects are primarily limited to boron-containing cancer cells. A phenylalanine derivative such as L-boronophenylalanine (L-BPA) was often used as a boron carrier to achieve preferential uptake of boron in tumor cells [6]. BNCT has been used for treatment in several case series of recurrent HNC, and both single-fraction and two-fraction BNCT had high response rates (60–83%) with acceptable toxicity levels [7,8]. Nuclear reactors have remained the most common neutron source for this treatment for decades.

National Tsing Hua University (NTHU) in Taiwan has a 2-MW TRIGA CONV research reactor (General Atomics, San Diego, CA, USA) named Tsing Hua Open-Pool Reactor (THOR), which is the only epithermal neutron source used for BNCT research. This reactor has an advantage depth of 8.9 cm, advantage ratio of 5.6, neutron flux of 1.69 × 10^19^ n/cm^2^/s, photon contamination of 1.25 × 10^−13^ Gy cm^2^, and fast neutron contamination of 2.8 × 10^−13^ Gy cm^2^ [9]. No clinical trial had been conducted on this research reactor until 2008, when a protocol involving a two-fraction design was drafted for treating recurrent HNC at our hospital. A two-fraction BNCT trial was conducted on 17 patients, and the results revealed that two-fraction BNCT had a high response rate and acceptable toxicity levels [10]. 

Despite initially promising treatment responses, further recurrence near sites re-irradiated with BNCT was common, according to our experience and that of other researchers [10,11]. We thus hypothesized that improved local control could be achieved through photon irradiation of a larger field around the recurrent gross tumor volume (GTV). Studies have employed image-guided intensity-modulated RT (IG-IMRT) as salvage therapy for recurrent HNC, achieving acceptable toxicity levels [12,13]. Accordingly, in 2013, we developed a new protocol combining IG-IMRT with BNCT. 

We conducted a trial to test the efficacy of the aforementioned protocol (i.e., IG-IMRT combined with BNCT) in increasing response rates without inducing excessive toxicity levels in HNC. The study was approved by the institutional review board of Taipei Veterans General Hospital in 2012 and by the Ministry of Health and Welfare of Taiwan in 2014, and it was registered at ClinicalTrials.gov (www.clinicaltrials.gov, accessed date: 31 December 2022). The trial was initiated in August 2014 and was terminated in 2022. This paper presents a report of the results of the trial on 14 patients with recurrent HNC who were treated with BNCT at a research reactor and with subsequent IG-IMRT at our hospital.

## 2. Materials and Methods

### 2.1. Study Design and Objectives

This was a single-center, prospective, noncomparative, open-label phase I/II trial for recurrent HNC. The primary endpoints were treatment toxicity and tumor response rates after single-fraction BNCT and fractionated IG-IMRT. The secondary objectives were the time required for tumor progression, progression-free survival (PFS), and overall survival (OS).

### 2.2. Inclusion and Exclusion Criteria

The following were the inclusion criteria for BNCT: (1) histologically proven malignancy of locoregionally recurrent tumors in the head and neck; (2) previous conventional RT already delivered for the disease (except melanoma); (3) bi-dimensionally measurable tumors evaluated with magnetic resonance imaging (MRI) or computed tomography (CT), with the largest tumor dimension being ≤12 cm; (4) aged between 18 and 80 years; (5) Eastern Cooperative Oncology Group (ECOG) performance status of ≤2; (6) white blood cell count of >2.5 × 10^9^/L, neutrophil count of >1.0 × 10^9^/L, platelet count of >75 × 10^9^/L, and serum creatinine level of <1.25 × upper limit of normal range; and (7) fluorine-18-labeled BPA positron emission tomography (^18^F-BPA-PET) demonstrated a BPA uptake T/N ratio of >2.5. The following were the exclusion criteria for BNCT: (1) histologically proven lymphoma or any other pathology type that was expected to respond to chemotherapy or to a safe dose of conventional RT; (2) an effective standard treatment option available; (3) distant metastasis outside the head and neck region; (4) a life expectancy of shorter than 3 months; (5) RT delivered less than 6 months before our trial; (6) high-dose RT (Biologically Effective Dose >70 Gy/35 fractions) already given for the present recurrent site within 1 year; (7) appearance of radiation myelitis or radiation necrosis of the brain or brain stem; (8) HNC surgery done within 6 months; (9) receiving concurrent systemic cancer therapy, including chemotherapy and targeted therapy; (10) having renal failure or severe congestive heart failure; (11) being pregnant; (12) being unable to sit or lie in a cast for 30–60 min; (13) a cardiac pacemaker installed or an unremovable metal implant in the head and neck region, which would interfere with dose planning or tumor response evaluation; (14) having carotid blowout syndrome with active bleeding within 6 months; or (15) having a history of malignancy other than HNC within 5 years (except for carcinoma in situ and nonmelanoma skin cancer). Furthermore, the inclusion criteria for IG-IMRT were as follows: (1) having an ECOG performance status of ≤2; (2) showing no evidence of disease progression during physical examination or CT simulation after BNCT; (3) already receiving nutrition support (feeding tube or intravenous fluid) for dysphagia, if present; and (4) having an improved or downgraded severity level of mucositis or radiation dermatitis compared with the severity level observed in the second or third week after BNCT. The exclusion criterion for IG-IMRT was as follows: having any grade 4 toxicity after BNCT.

### 2.3. Imaging Studies

Before the screening procedure, each patient underwent MRI for evaluating the size of and staging the recurrent tumor. At our hospital, a GE VCT scanner (GE Medical Systems, Milwaukee, WI, USA) is used for PET/CT scanning. In this trial, whole-body ^18^F-fluorodeoxyglucose (FDG)-PET/CT was conducted to rule out distant metastases. The ^18^F-BPA-PET/CT procedure was repeated before BNCT to derive T/N ratios. The injected dose of ^18^F-BPA was approximately 5 mCi. ^18^F-BPA uptake in the tumor was measured by calculating the mean standard uptake value (SUV) for the GTV. ^18^F-BPA uptake in the normal tissue was measured by calculating the mean SUV for a region in the subcutaneous connective tissue on the contralateral side of the GTV; this measurement was conducted 60 min after ^18^F-BPA injection. 

### 2.4. Treatment Planning for BNCT

A CT simulation before BNCT was routinely conducted in the supine position by using a slice thickness of 5 mm. To delineate the GTV (including the primary tumor and adjacent lymphadenopathy if present), CT images obtained from the above procedure were co-registered with T1-weighted MRI and ^18^F-BPA-PET images. Normal structures (including critical organs, such as the eyes, optic nerves, lens, brain stem, inner ears, parotid glands, spinal cord, mandible, and oral mucosa) were delineated. The CT images and contours of the participants were subsequently sent to the university for processing. A treatment-planning software named THORplan [14] was utilized to perform dose calculations, display three-dimensional dose distributions on CT images, and produce dose volume histograms for both tumors and normal structures. Tissue compositions were defined in accordance with the recommendations of the International Commission on Radiation Units and Measurements [15]. The relative biological effectiveness (RBE) was 1 for gamma rays and 3.2 for neutrons. The compound biological effectiveness (CBE) for boron was 4.9 for the mucosa, 3.8 for the tumor, 2.5 for the skin, and 1.3 for other normal tissues [16]. 

Both the physical dose and biologically equivalent dose were calculated during the BNCT planning process. The equivalent dose (D-Eq) was generated as the sum of the physical dose components multiplied by the weighting factors (including the RBE and CBE) of each dose component in a tissue. We practiced the following guidelines of dose prescription: (1) delivering 20–25 Gy-Eq/fraction as the average dose of the GTV; (2) limiting the volume of oral mucosa receiving >10 Gy-Eq/fraction to the lowest possible level; and (3) limiting the maximum dose available to the optic nerve or chiasma to <8 Gy-Eq/fraction. The dose constraints for other critical organs like the spinal cord and brain stem were determined individually according to the previously received radiation dose. A single anterior, lateral, or anterior–oblique portal was used for all but one participant. One patient was treated with two opposing lateral fields. 

### 2.5. Treatment Planning for IG-IMRT

CT simulations were repeated at least 3 weeks after BNCT with slice thickness of 3 mm; for these simulations, each patient was immobilized using a thermoplastic immobilization device (mask), and the simulations involved medium contrast enhancement. Treatment planning for IG-IMRT was conducted using Velocity software (Accuray Inc. Sunnyvale, CA, USA). The GTV was recontoured using the images obtained from the second CT simulations. Subsequently, a selected margin (3–5 mm) was added around the GTV to generate the clinical target volume (CTV), and an additional 3 mm margin was added around the CTV to generate the planning target volume (PTV). If not previously irradiated, lymphatic drainage areas adjacent to the GTV were included in the CTV. Furthermore, additional care was taken to limit the dose to critical organs such as the carotid artery, spinal cord, brain stem, and mandible. The planned dose/fractionation for the PTV in IG-IMRT was 45 Gy/25 fractions, 5 fractions/week.

### 2.6. BNCT and IG-IMRT Administration

L-BPA (Hammercap AB, Sweden and Taiwan Biopharm Company, Taiwan) was complexed with fructose to form an L-BPA–F solution to increase its solubility [17]. It was stored at a concentration of 25 g/L and a pH of 7.6 [18]. An on-site simulation placing patients in a stable posture (sitting or lying) during exposure to a horizontal radiation field with polyethylene extension collimators over the beam aperture was conducted at the research reactor. The tumor location and irradiation volume (GTV with a margin of at least 1 cm) were verified through a beam’s eye view. The intravenous L-BPA–F (500 mg/kg) infusion was administered in two phases: the first phase involved drug infusion at a rate of 200 mg/kg/h for 2 h before neutron irradiation; the second phase involved drug infusion at a rate of 100 mg/kg/h along with irradiation, and the process was stopped when the beam was off. Six blood boron concentration measurements using inductively coupled plasma atomic emission spectrometry were done before, during, and after injection. The blood boron concentration during BNCT was estimated with the measurements immediately before and after irradiation. Daily fractionated IG-IMRT was executed using a tomotherapy Hi-ART system (Accuray, Sunnyvale, CA, USA) and was initiated 4 weeks after BNCT if not contraindicated.

### 2.7. Follow-Up

After BNCT, the patients were assessed weekly for 4 weeks to monitor for any adverse effects. Each assessment included a physical examination; weight and vital sign measurements; and performance status, general well-being, and toxicity assessments. After IG-IMRT, the patients were followed up every 3 months for the first 2 years and at least every 6 months for the subsequent 3 years. Tumor responses were evaluated through a physical examination and MRI conducted at least 1 month after the final fraction of IG-IMRT. ^18^F-FDG-PET/CT was conducted at least 3 months after the last BNCT to confirm a complete response (CR) and rule out distant metastases. 

### 2.8. Criteria for Response and Toxicity Evaluation

Tumor responses were assessed in accordance with the Response Evaluation Criteria in Solid Tumors (RECIST)v1.1. In addition, adverse effects were assessed in accordance with the Common Terminology Criteria for Adverse Events v3.0. 

### 2.9. Statistical Analysis

The efficacy of the treatment protocol was analyzed on the basis of the intent-to-treat principle. The analysis included all patients who received at least BNCT. OS was calculated from the date of BNCT to the date of death by using the Kaplan–Meier method. Patients who were alive at the analysis cut-off date (30 Septcember 2022) were censored. Locoregional PFS was calculated from the date of the first BNCT session to the date of locoregional progression. Patients who were alive or dead and had no evidence of locoregional progression were also censored. Log rank test was used to compare survivals from different trials. All statistical analyses were performed using Stata 12.0 (StataCorp. 2011, College Station, TX, USA).

## 3. Results

### 3.1. Participant Accrual and Tumor Characteristics

This trial enrolled 14 patients with a total of 18 tumors between July 2014 and September 2021. The most common primary tumor site was the oral cavity (50%). All patients had previously received at least one course of photon RT. The demographic and tumor characteristics of the patients are listed in Table 1. All 14 patients received ^18^F-BPA-PET/CT before BNCT. 

### 3.2. Radiation Dose Delivered

The median blood boron concentration was 25.9 ppm (range: 22–29.7 ppm) during irradiation. The median neutron irradiation time was 21.1 min (range: 16.1–27.0 min). For the GTV, the median average dose in BNCT was 21.6 Gy-Eq (range: 10.7–32.3 Gy-Eq), and the median prescription dose in IG-IMRT was 46.8 Gy (range: 41.4–53). The treatment parameters are listed in Table 2.

### 3.3. Efficacy

The median follow-up period was 11.8 months (range: 3.6–53.2 months). Five patients had a CR (verified through PET 3 months after IG-IMRT), and four had a partial response. The dose distribution and imaging results obtained for two patients before and after BNCT + IG-IMRT treatment are presented in Figure 1 and Figure 2, respectively. One of these patients remained disease free for 2 years after BNCT. The median time for locoregional tumor progression was 8.3 months (range: 1.3–25.9 months). The 1-year local PFS for all patients was 21% (95% confidence interval (CI): 5–45%; survival curves in Figure 3), and the 1-year OS for all patients was 56% (95% CI: 3–77%; survival curves in Figure 4). Patients who received both BNCT and IMRT had a 1-year local PFS of 25% (95% CI: 6–50%). 

### 3.4. Toxicity

The acute and late toxicity levels observed after treatment are listed in Table 3 and Table 4. The most common low-grade acute toxicities were mucositis, dermatitis, and alopecia (100%). Two patients with a tumor volume of >100 cc (total of three cases) developed grade 3 infections or tumor pain, and one patient developed grade 4 dyspnea due to laryngeal edema. Another patient had a grade 4 hemorrhage, which was salvaged through arterial embolization; the patient was alive and experienced no disease recurrence 26 months after BNCT. The most common grade 3 late toxicity was skin ulceration (four cases). Grade 1 and 2 chronic pain was also common and was managed at our outpatient department. One grade 4 late hemorrhage was observed, and the patient recovered after intensive care. No grade 5 late toxicity was observed.

### 3.5. Failure Pattern

All failures occurred within the head and neck region. All recurrence sites, except for one, were within or adjacent (<1 cm) to our radiation fields. Two patients had recurrence on the side opposite to our treatment fields, and one of them also had a failure in the combined treatment area. No patient presented with lymph node or distant metastases. 

### 3.6. Other Therapies Administered after Combined Treatment or BNCT Alone

Among the patients with disease progression after BNCT, five received salvage chemotherapy, three received surgery for local re-recurrence, and two received immunotherapy. One patient received further compassionate BNCT outside this trial.

## 4. Discussion

Despite advances in modern RT techniques, reirradiation in locally advanced HNC carries a risk of severe complications. BNCT, a type of targeted RT, can deliver a high dose of radiation to the tumor while limiting the dose to the surrounding normal tissues to the lowest possible level. Clinicians have employed tomotherapy for salvage RT in locally recurrent HNC [12,13]. According to our review of the literature, the current trial is the first to combine BNCT and IG-IMRT photon therapy for locally recurrent HNC. 

The disadvantages of using BNCT as a single treatment include the following: (1) the BPA distribution in the tumor is not homogeneous, particularly in large tumors with hypoxic areas; (2) the dose or coverage is insufficient for the CTV, which is difficult to define in reirradiation situations; and (3) the penetration of epithermal neutrons for deep-seated tumors is impaired. Our combined therapeutic approach has the following theoretical advantages: (1) it enables adaptive planning with IMRT after BNCT; (2) it compensates for the dose inhomogeneity of BNCT and provides a higher dose to the CTV through IMRT [19]; (3) it enables elective nodal irradiation through RT for patients with a high risk of neck metastasis (such elective nodal irradiation was administered to two patients); and (4) it enables precise reirradiation through IG-IMRT to spare the carotid artery and other critical organs. The disadvantages of our combined approach include the following: (1) multi-fraction IMRT has a longer treatment time compared with the time required for the second fraction of BNCT in the previous protocol, and (2) toxicity levels to normal structures or tissues may be increased due to a greater radiation volume. However, this study did not detect any grade 5 toxicity. 

We used a relatively small CTV margin (3 to 5 mm) around the GTV in the IMRT process to avoid severe complications. This margin was probably not sufficiently wide for some patients, as evidenced by the high incidence of further recurrence around the CTV. A wider margin (i.e., 1 cm or larger) might prevent further recurrence around the GTV, particularly for tumors of a smaller volume (<50 cc) and those at a sufficient distance from critical organs (e.g., the brain stem, spinal cord, and carotid artery). Two patients had recurrence on the opposite side of the treatment field; such contralateral recurrence was not related to the size of the CTV margin and was generally not treated during reirradiation. Only two patients received prophylactic adjacent neck irradiation because these regions were not included in the previous treatment. This study did not record any nodal failure.

The optimal BNCT and IMRT doses remain to be determined. In this trial, we empirically prescribed the BNCT dose based on data obtained from our previous two-fraction BNCT trial. In common practice, patients with subclinical disease generally received IMRT at a dose of 45 to 50 Gy/25 fractions. This dose might be decreased when combined with BNCT. The present trial included a 4-week gap between the BNCT and IMRT treatments to allow for recovery from acute mucositis and dermatitis induced by the first treatment. The accelerated tumor cell repopulation may occur during this rest period, which may impair the efficacy of subsequent IMRT. Nevertheless, the second CT simulation did not indicate any increase in GTV 3 weeks after BNCT. 

The present trial was noted to have a similar response rate (64%) to our previous two-fraction BNCT trial (71%). There was no difference in the local PFS (*p* = 0.274) or OS (*p* = 0.291) of these two by log rank test. However, the 2-year PFS of the patients receiving the combined treatment in the present trial (8%) was inferior to that of those in the previous trial (28%) and the Finish trial (20%) [11]. A recent Japanese study with cyclotron-based BNCT for recurrent HNC reported a median local/regional PFS of 11.5 months [20], which was longer than this trial (8.3 months). Since most of the failure was in-field or marginal recurrence in the current study, the local control was not improved by the combined treatment strategy with extra radiation dose delivered to a larger CTV or better tumor dose homogeneity. In fact, a fair comparison between our trial or with other trials is difficult because all were phase II non-randomized trials conducted during different periods. In addition, the pathologies, tumor sites, and clinical stages of the patients in the present trial differed from those of the patients in the previous trials. As expected, both our trials revealed a high CR rate (67% for the previous and 100% for the current trial) in patients with nasopharyngeal cancer, which is more radiosensitive than tumors from other sites in the head and neck region.

For patients with a GTV larger than 100 cc, our combined therapeutic approach may increase the chance of severe toxicity owing to the large irradiation volumes, as exemplified by two patients in this trial, one of whom developed a grade 3 infection, and the other developed grade 4 dyspnea. Fractionated or single-fraction compassionate BNCT alone with reduced dose per fraction may represent an appropriate choice for avoiding grade 4 toxicity in this scenario. Prophylactic stent placement before salvage BNCT may have prevented carotid blowout in two patients with tumors abutting the artery. Our patients commonly experienced late toxicity in the form of skin ulcers in the cheek and bone necrosis in the mandible, particularly those with oral cavity cancer. Therefore, clinicians should carefully perform combined BNCT and IG-IMRT on patients who have received high-dose photon RT in the oral cavity/oropharynx before recurrence. If long-term tumor control is possible, reconstructive plastic surgery may help improve patients’ quality of life. One patient in our study underwent this procedure. 

The present study has the following limitations: First, the sample size was small, and the heterogeneity of tumors was high. Therefore, analyzing prognostic factors was difficult. Second, the follow-up period was not sufficiently long. Third, the dosimetry for BNCT and combined IG-IMRT involved more complexity and uncertainty compared with that for pure photon beam treatment planning [5,21]. Finally, the dose and volume of previously received photon irradiation before recurrence may have influenced the chosen dose and volume of CTV for IMRT and may have thus affected the toxicity and PFS in the current trial. 

There have been challenges to the clinical trials programs of BNCT in our country. Our clinical trials with BNCT started in 2010, and the first one was designed only for recurrent HNC because this cancer is an endemic disease in Taiwan. THOR was the only epithermal neutron source, and it was maintained by the National Tsing Hua University. Funding is a problem for BNCT trials, including the current study. Therefore, there are limited types of disease that could be the target of BNCT trials. Furthermore, THOR, a research reactor, is not a dedicated medical device and the maximal number of patients that could be treated in one working day is only two up to now. On the other hand, we started compassionate BNCT in 2017, and the case number treated at THOR is increasing every year. Besides HNC, brain tumors have been treated there since then. Clinical trials for recurrent brain tumors are now drafted, and an accelerator-based BNCT center is under construction. In the future, we expect to see more BNCT clinical trials done with both the reactor and accelerator in our country.

In conclusion, clinicians must exercise caution when administering combined BNCT and IMRT for locally recurrent HNC, particularly for large tumors (>100 cc), for tumors abutting the carotid artery, and/or for patients with oral cavity cancer. Individual dose modification should be employed to achieve a balance between toxicity and local control. This study recorded a high incidence of in-field and marginal recurrence despite the moderately enlarged CTVs covered by IMRT. Future studies should consider further revising our protocol; for example, they could consider combining BNCT with additional modalities other than radiation to procure better results.

## Figures and Tables

**Figure 1 cancers-15-02762-f001:**
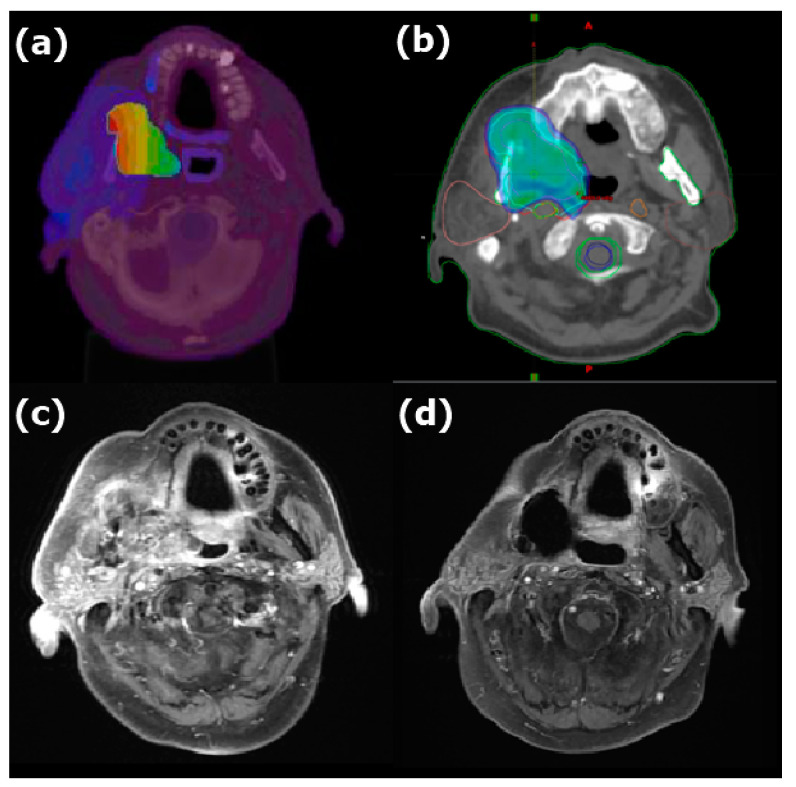
Dose distribution of (**a**) BNCT and (**b**) IMRT and MRI (**c**) before and (**d**) 11 months after combined treatment for recurrent oral cavity carcinoma. The color wash in (**a**) showed the BNCT dose level represented by a spectrum from the blue to the red (3.5 to 35.1 Gy-Eq, respectively). The color wash in (**b**) showed the area covered by 45 Gy and above with IMRT. The mean dose of BNCT was 23.0 Gy-Eq in a single fraction, and the prescription dose of IMRT was 45 Gy/25 fractions. The tumor disappeared completely after combined treatment.

**Figure 2 cancers-15-02762-f002:**
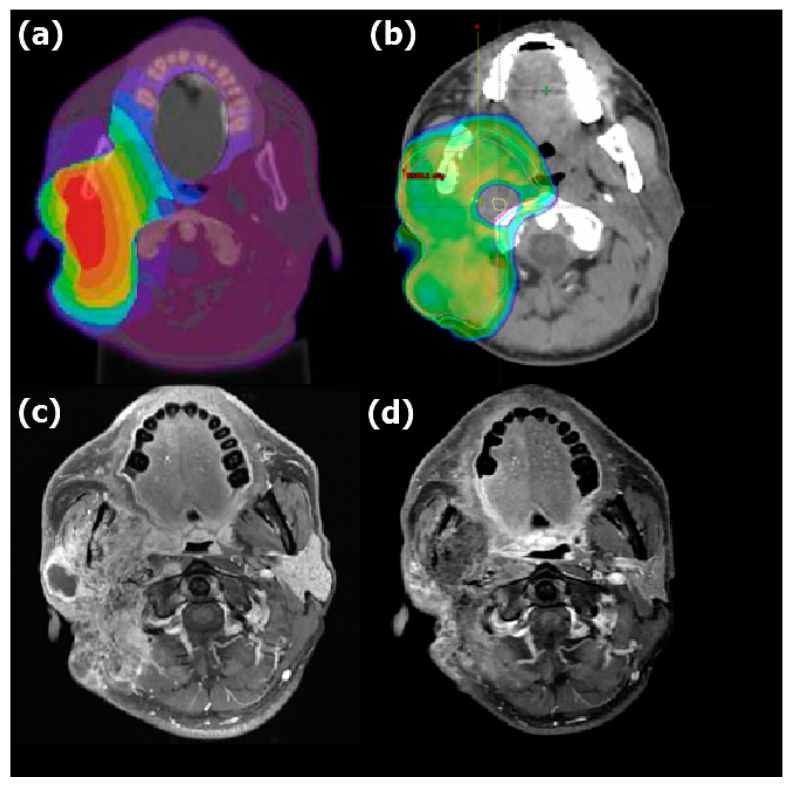
Dose distribution of (**a**) BNCT and (**b**) IMRT and MRI (**c**) before and (**d**) 6 months after combined treatment for recurrent mucoepidermoid carcinoma of the right parotid gland. The color wash in (**a**) showed the BNCT dose level represented by a spectrum from the blue to the red (3.8 to 38.0 Gy-Eq, respectively). The color wash in (**b**) showed the area covered by 45 Gy and above with IMRT. The mean dose of BNCT was 25.4 Gy-Eq in a single fraction, and the prescription dose of IMRT was 45 Gy/25 fractions. The tumor decreased in size significantly after therapy.

**Figure 3 cancers-15-02762-f003:**
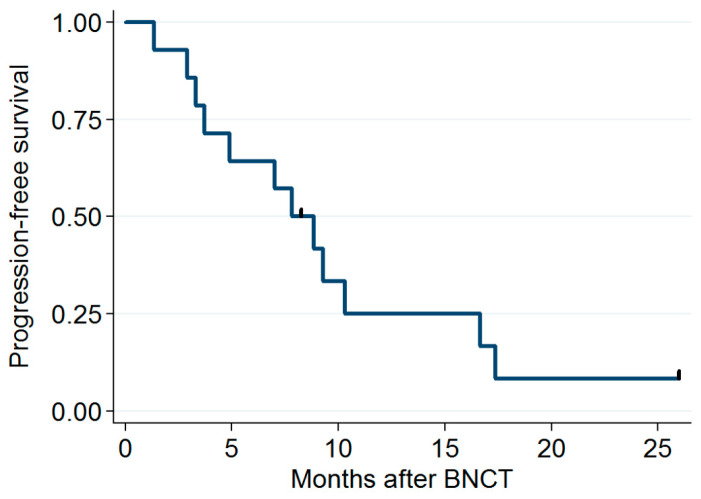
Progression-free survival rates.

**Figure 4 cancers-15-02762-f004:**
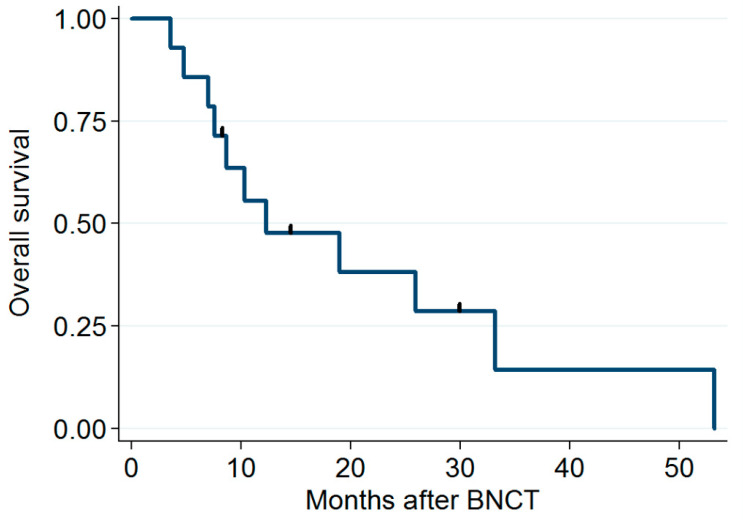
Overall survival rates.

**Table 1 cancers-15-02762-t001:** Patient demographic and tumor characteristics.

Characteristics	*n* (%)	Median (Range)
*Patient*		
Gender		
Male	10 (71)	
Female	4 (29)	
Age (y)		57 (40, 78)
Surgery before BNCT		
Yes	8 (57)	
No	6 (43)	
Prior cumulative RT dose (Gy)		68 (60, 102)
Time from prior RT to BNCT (mo)		23 (8, 108)
*Tumor*		
Primary site		
Oral cavity	7 (50)	
Nasopharynx	2 (14)	
Oropharynx	2 (14)	
Hypopharynx	1 (7)	
Parotid gland	1 (7)	
Mandible	1 (7)	
Pathology		
SCC	10 (71)	
Other carcinoma *	3 (21)	
Sarcoma	1 (7)	
Volume (cc)		13.5 (3.1, 368)
Recurrent T stage		
rT0	1 (7)	
rT1	0 (0)	
rT2	1 (7)	
rT3	6 (43)	
rT4	6 (43)	
Recurrent N stage		
rN0	9 (64)	
rN1	2 (14)	
rN2	3 (21)	
rN3	0 (0)	

RT, radiotherapy; BNCT, boron neutron capture therapy; mo, months; SCC, squamous cell carcinoma. * Other carcinoma includes mucoepidermoid carcinoma and non-keratinizing carcinoma.

**Table 2 cancers-15-02762-t002:** BNCT and IG-IMRT parameters.

Parameters	Median (Minimum, Maximum)
BNCT	
BPA T/N ratio by PET	3.0 (2.5,3.6)
Irradiation time (min)	21.0 (16.7, 27.0)
Average equivalent tumor dose (Gy (Eq))	21.6 (10.7, 32.3)
Average tumor physical dose (Gy)	6.1 (3.4, 9.8)
Maximum mucosa physical dose (Gy)	4.9 (3.0, 6.1)
Interval between BNCT and IG-IMRT (days)	30 (26, 39)
IG-IMRT	
Total prescription dose	46.8 (41.4,53)
Total fractions	25 (23, 28)

**Table 3 cancers-15-02762-t003:** Acute * toxicities.

Adverse Effects	None (*n*)	Grades 1–2 (n)	Grade 3 (*n*)	Grade 4 (*n*)	Not Available
Mucositis	0	14	0	0	0
Radiation dermatitis	0	14	0	0	0
Alopecia	0	14	0	0	0
Dysphagia	3	6	2	0	3
Tumor pain	3	8	3	0	0
Hemorrhage	10	3	0	1	0
Infection (soft tissue)	5	6	2	0	1
Edema (H&N)	10	3	1	0	0
Edema (laryngeal)	13	0	0	1	0
Otitis	12	2	0	0	0
Nausea or vomiting	9	5	0	0	0
Trismus	12	2	0	0	0

H&N, head and neck. * Toxicities that developed within 3 months after the last fraction of IG-IMRT.

**Table 4 cancers-15-02762-t004:** Late * toxicities (evaluable in 12 cases).

Adverse Effects	None (*n*)	Grades 1–2 (*n*)	Grade 3 (*n*)	Grade 4 (*n*)
Osteoradionecrosis	8	3	1	0
Soft tissue necrosis	4	6	2	0
Hemorrhage	11	0	0	1
Cranial neuropathy	9	1	2	0
Dysphagia	6	5	1	0
Pain	2	8	2	0
Skin ulceration	7	1	4	0
Alopecia	12	0	0	0
Impaired hearing	11	1	0	0
Otitis	9	2	1	0
Xerostomia	8	4	0	0
Edema (H&N)	10	2	0	0
Trismus	8	3	1	0
Infection	4	7	1	0

H&N, head and neck. * Toxicities that developed beyond 3 months after the last fraction of BNCT.

## Data Availability

The datasets used and/or analyzed during the current study are available from the corresponding author on reasonable request.

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
