# Peer review of "Boron Neutron Capture Therapy Followed by Image-Guided Intensity-Modulated Radiotherapy for Locally Recurrent Head and Neck Cancer: A Prospective Phase I/II Trial"

_cancers, 2023, doi:10.3390/cancers15102762_

Round 1

Reviewer 1 Report

The authors investigated the safety and efficacy of BNCT followed by image-guided IMRT for recurrent H&N cancer.

This is novel clinical research and could be done only by the authors team even in the world.

Therefore, this is worthy for publication in Cancers focused on BNCT.

Prior to final decision, this reviewer would like to confirm the following items.

1.     How did the authors decide the sample size?

2.     The authors should compare the current study to the previous their own 2-times BNCT for the same disease with statistical analysis. Also they should analyzed the current data with other BNCT clinical studies and trials such as reactor-based BNCT by Kankaanranta et al. and accelerator-based BNCT by Hirose, et al with statistical analysis.

3.     The authors stated the rationale of this 2-staged strategy for recurrent H&N cancers in lines 293 to 307. Could the authors elicite any conclusions or legitimacy of their hypothesis from current study, finally?  

4.     With my knowledge, BNCT showed good local response but could not show the inhibition of remote recurrence for advanced H&N cancers. However in their series, there was no remote recurrence, why?

Author Response

Thanks for your comments. Listed below are my responses:

  1. In the protocol we used Simon’s optimal two-stage test (alpha=0.05, beta=0.1, power=90 %, one-sided test) to determine the sample size. We estimated that BNCT combined with IG-IMRT for recurrence is considered worth further study if the response rate (including complete and partial responses) is 60 % or higher (P1) and reject the treatment if response rate is 30 % or lower (P0). In the first stage 10 patients will be enrolled, if more than 3 patients have response, then another 18 patients will be enrolled. The test regimen will be considered effective if 13 or more patients have response in the second stage.  A maximal sample size of 28 is required. However, the patient enrollment was slow and we ran out of funding after 9 patients treated with BNCT and 7 received combined treatment. The 2nd funding source could provide a quota of only 5 more participants .  Therefore, we terminated the study at a sample size of 14.
  2.  Compared with our previous 2-fraction BNCT trial, there was no difference in the local progression-free or overall survival by log rank test(p=0.274 and 0.291 respectively). We add the survival comparison data in the discussion section(line 342 to 347). Since we don't have the original survival data of the Finish or Japanese study, head-to-head comparison with statistical methods is impossible.
  3. The hypothesis of "improved local control could be achieved through photon irradiation of a larger field around the recurrent gross tumor volume (GTV)" of this trial was not verified by the current study.  We added the above statements in the discussion section (line 347 to 350) and re-emphasized it in the conclusion(line 395 to 396).
  4. Generally speaking, head and neck cancer had more local or regional (neck) failure than distant metastasis perhaps due to their biological characteristics. Local treatment like surgery and radiation(including BNCT) play important role in disease control. Many patients died of local/regional recurrence before distant failure developed.

Reviewer 2 Report

The heterogeneity of tumors and uneven distribution of boron in tumor tissue are considered to be the reasons for local recurrence or tumor progression after BNCT. Therefore, supplementing the tumor area or around with a certain dose of X-ray after BNCT may improve the overall control rate of the tumor. Japanese researchers have made good attempts on the "Operation+BNCT+IMRT" treatment plan for malignant glioma treatment. This article first uses the BNCT+(IG-IMRT) method in the treatment of recurrent head and neck tumors. This article clearly describes the parameters, process, and results of this clinical trial, which has high reference value for future BNCT clinical plans for recurrent head and neck cancer. I agree with the publication of this article. The following aspects are some suggestions to make the reader known better of this work.:

1.Neutron beam parameters are important for the procedure of BNCT. It is better to add some descriptions of THOR epithermal beam in the Materials and Methods section.

2. For the BNCT dose distribution in Figures 1a and 2a, as well as the IMRT dose distribution in Figures 1b and 2b, please add legends or text to explain the dose range represented by each color or curve.

3. In this article, it is mentioned that the average dose of GTV in BNCT is 21.6 Gy-Eq, the average total prescription dose of IG-IMRT is 46.8 Gy, and the time interval between BNCT and IG-IMRT is 30 days. Could there be more discussion on the selection of these important treatment parameters, such as the basis for selecting these parameters?

Author Response

Thanks for your comments. Listed below are my responses:

  1. Neutron beam parameters are mentioned in line 73 to 75 and a new reference is added(refernce 9).
  2. We added sentences in the figure legends for Fig. 1 (line 255 to 257)and 2  (line 262 to 265)to explain the dose range represented by the color wash shown in these figures.
  3. Yes. From line 332 to 338, we explained the rationale behind the dose selection for BNCT and IMRT. It is derived from our experience of the previous BNCT trial and common clinical practice. The actual delivered IMRT median dose was higher than the planned dose in the protocol(line 184) due to dose modification in individual cases.  We assumed that IMRT with 45 Gy/25 fx one month after the first fraction BNCT could procure better tumor response and local control than those from our previous 2-fraction BNCT trial (dose of the first fraction and time interval were the same for both trials).   

Round 2

Reviewer 1 Report

The authors revised well according to the revier's suggestion.